# Global sensitivity analysis of chemistry-climate model budgets of tropospheric ozone and OH: Exploring model diversity

Oliver Wild[1], Apostolos Voulgarakis[2], Fiona O'Connor[3], Jean-François Lamarque[4], Edmund M. Ryan[1,5], and Lindsay Lee[6,7]

[1]Lancaster Environment Centre, Lancaster University, Lancaster, UK
[2]Department of Physics, Imperial College, London, UK
[3]Met Office Hadley Centre, Exeter, UK
[4]National Center for Atmospheric Research, Boulder, CO, USA
[5]Now at: School of Mathematics, University of Manchester, Manchester, UK
[6]School of Earth and Environment, University of Leeds, Leeds, UK
[7]Now at: Department of Engineering and Mathematics, Sheffield Hallam University, Sheffield, UK

**Correspondence:** Oliver Wild (o.wild@lancaster.ac.uk)

**Abstract.** Projections of future atmospheric composition change and its impacts on air quality and climate depend heavily on chemistry-climate models that allow us to investigate the effects of changing emissions and meteorology. These models are imperfect as they rely on our understanding of the chemical, physical and dynamical processes governing atmospheric composition, on the approximations needed to represent these numerically, and on the limitations of the observations required to constrain them. Model intercomparison studies show substantial diversity in results that reflect underlying uncertainties, but little progress has been made in explaining the causes of this or in identifying the weaknesses in process understanding or representation that could lead to improved models and to better scientific understanding. Global sensitivity analysis provides a valuable method of identifying and quantifying the main causes of diversity in current models. For the first time, we apply Gaussian process emulation with three independent global chemistry transport models to quantify the sensitivity of ozone and hydroxyl radicals (OH) to important climate-relevant variables, poorly-characterized processes and uncertain emissions. We show a clear sensitivity of tropospheric ozone to atmospheric humidity and precursor emissions which is similar for the models, but find large differences between models for methane lifetime, highlighting substantial differences in the sensitivity of OH to primary and secondary production. This approach allows us to identify key areas where model improvements are required while providing valuable new insight into the processes driving tropospheric composition change.

*Copyright statement.* TEXT

## 1 Introduction

Atmospheric photochemistry and transport processes play important roles in the Earth system by controlling the impact of natural and anthropogenic trace gas emissions on air quality and global climate. Methane ($CH_4$) and ozone ($O_3$) are the

second and third most important greenhouse gases contributing to climate change since the preindustrial era (IPCC, 2013). The atmospheric abundance of both gases has increased substantially due to anthropogenic activity, and their fates are strongly coupled through the short-lived hydroxyl (OH) radical. $CH_4$ is an $O_3$ precursor and $O_3$ is a major source of OH, which controls the oxidation of $CH_4$ and many other trace gases. At the surface $O_3$ contributes to poor air quality and is damaging to human health, crop yields and natural ecosystems (Monks et al., 2015). The relatively short lifetime of these gases makes them attractive targets for emission controls (Shindell et al., 2012), but scientific uncertainties associated with the processes that govern their abundance and distribution has hindered implementation of effective control policies.

Current global chemistry-climate models representing the co-evolution of atmospheric $O_3$ and $CH_4$ show differences in $CH_4$ lifetime of almost a factor of two (Wild, 2007; Voulgarakis et al., 2013). This prevents them from simulating the observed atmospheric build-up of $CH_4$ correctly or attributing its causes reliably, and leads to substantial uncertainty in the impact of future emission changes on global climate (Stevenson et al., 2013; IPCC, 2013). The underlying cause is differences in OH, which depends on humidity, sunlight, $O_3$, and on a wide range of chemical and dynamical processes. For $O_3$, on the other hand, the abundance, seasonality and spatial variation are represented relatively well in models under present-day conditions, but observed changes in surface $O_3$ since the preindustrial era are thought to be underestimated (Stevenson et al., 2013), although there is continuing uncertainty surrounding preindustrial levels (Tarasick et al., 2019). Models have difficulty reproducing recent observed trends in surface $O_3$ driven by changes in precursor emissions, natural sources, stratospheric influx and transport patterns (Parrish et al., 2014). This is a major concern because changes in the tropospheric abundance of $O_3$ influence our assessment of radiative forcing and also attainment of air quality objectives on regional and urban scales (e.g., Akimoto, 2003). These discrepancies suggest that there are major weaknesses in our fundamental understanding of the chemical, dynamical, and emission processes controlling the distribution, interaction and fate of $O_3$, $CH_4$ and OH, or in how these processes are represented in global chemistry and climate models.

Global sensitivity analysis provides a valuable approach to determine the major drivers of model behaviour, and has been applied to atmospheric chemistry schemes to explore uncertainties in tropospheric $O_3$ (Derwent and Murrells, 2013; Christian et al., 2017; Ridley et al., 2017; Newsome and Evans, 2017). These studies have typically used Monte Carlo-based ensemble approaches for simple models (e.g., Ridley et al., 2017) or structured random-sampling approaches for more computationally intensive models (e.g., Christian et al., 2017), and have focussed on sensitivities in a single model framework. In this study we demonstrate the use of Gaussian process emulation for global sensitivity analysis, applied previously to models of aerosol processes (Lee et al., 2011, 2013) and air quality (Beddows et al., 2017; Aleksankina et al., 2019), and apply it to explore the sensitivity of global tropospheric $O_3$ and $CH_4$ lifetime to uncertainty in key model processes and inputs. We investigate how the sensitivities differ across three independent chemistry-transport models, and demonstrate how this approach may be used to explore the diversity in model responses and to identify where model results differ.

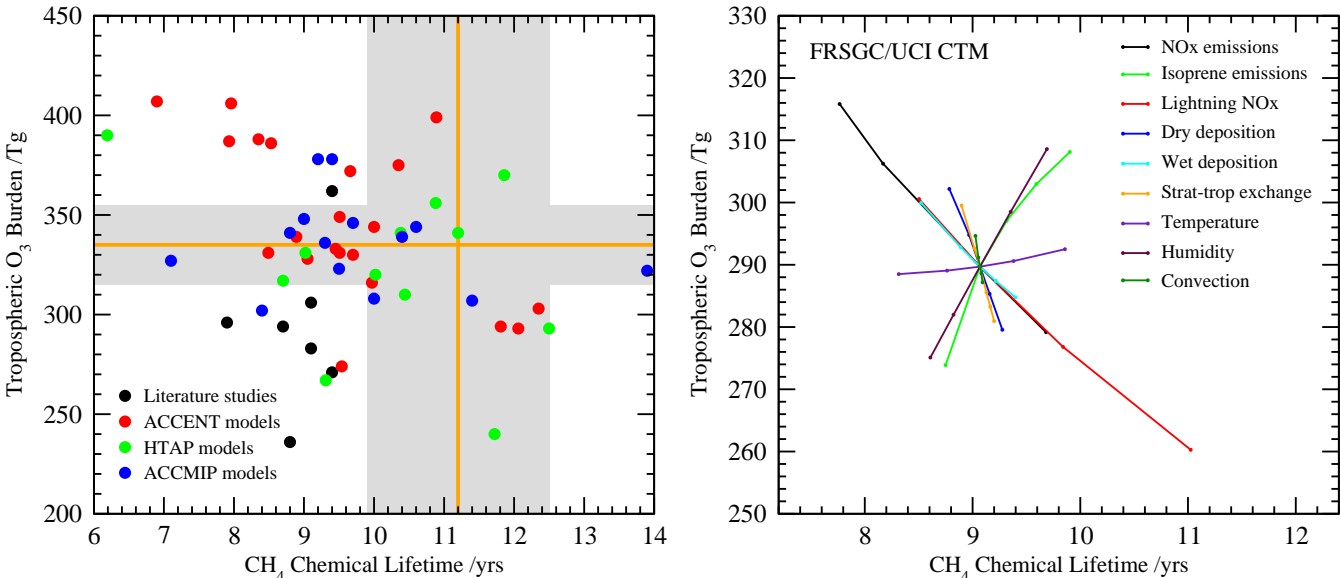

**Figure 1.** Tropospheric oxidant budgets from previous published studies and model intercomparisons (left panel, **a**), along with measurement-based estimates of the tropospheric $O_3$ burden and $CH_4$ lifetime (shaded regions). The right panel **(b)** shows results from one-at-a-time sensitivity studies with a single model revealing the extent to which individual processes can influence the budgets (see Wild (2007) for details). Note that results in the left panel represent differing emissions and meteorological years (study details are given in Table 1), and that the right panel covers only part of the parameter space shown in the left panel.

## 2   Approach

We consider here two important global diagnostics of model performance, the tropospheric $O_3$ burden and the chemical lifetime of $CH_4$ in the troposphere. The tropospheric $O_3$ burden is the annual mean mass of $O_3$ below the tropopause, defined here by the 150 ppb isopleth of monthly mean $O_3$. The chemical lifetime of $CH_4$ reflects the lifetime of $CH_4$ to removal by OH in the troposphere, and provides a useful proxy for global tropospheric oxidizing capacity. Global model studies in the literature and previous model intercomparisons show a large diversity in modelled budgets (see Fig. 1), where the range in $O_3$ burden and $CH_4$ lifetime both span about a factor of two. There is no clear relationship between the budget terms on an annual basis, highlighting the relatively complex relationship between tropospheric $O_3$ and OH that reflects physical and dynamical processes as well as photochemistry.

Observation-based determination of these global quantities is difficult. However, assessment of three global $O_3$ climatologies derived from ozonesonde measurements over the 1980s and 1990s indicates an annual mean tropospheric $O_3$ burden of 327–344 Tg when applying the same 150 ppb isopleth definition of the tropopause used in model analysis (Wild, 2007), suggesting a burden of about 335±20 Tg. Recent satellite and ozonesonde-based estimates of the global burden range from 333–345 Tg (Gaudel et al., 2018). Ensemble mean $O_3$ burdens from recent model intercomparisons lie close to this: 344±39 Tg from ACCENT (Stevenson et al., 2006), 328±41 Tg from HTAP (Fiore et al., 2009) and 337±23 Tg from ACCMIP (Young et

**Table 1.** Global tropospheric metrics from previous model studies

| Studies | Number | $O_3$ burden | $CH_4$ lifetime | References |
|---------|--------|------------|---------------|------------|
| Early literature studies | 33 studies | 307±38 Tg | | Wild (2007) |
| ACCENT intercomparison | 21 models | 344±39 Tg | 9.6±1.4 yr | Stevenson et al. (2006) |
| HTAP intercomparison | 12 models | 328±41 Tg | 10.2±1.7 yr | Fiore et al. (2009) |
| ACCMIP intercomparison | 14 models | 337±23 Tg | 9.8±1.6 yr | Young et al. (2013); Voulgarakis et al. (2013) |
| Observational estimates | | 335±20 Tg | 11.2±1.3 yr | Wild (2007); Prather et al. (2012) |

al., 2013), see Table 1, but about half of published studies lie outside the observationally-constrained range (see Fig. 1). A thorough observation-based sensitivity analysis of the factors contributing to $CH_4$ removal gave a whole-atmosphere lifetime of 9.1±0.9 yr, and a corresponding $CH_4$ chemical lifetime of 11.2±1.3 yr (Prather et al., 2012). The latter is substantially longer than that derived from model intercomparisons: 9.6±1.4 yr from ACCENT (Stevenson et al., 2006), 10.2±1.7 yr from HTAP (Fiore et al., 2009) and 9.8±1.6 yr from ACCMIP (Voulgarakis et al., 2013), and two thirds of the model studies shown in Fig. 1 lie outside this range. However, it is difficult to judge the validity of existing model results without a clearer idea of the uncertainties involved and how they contribute to the corresponding biases.

The sensitivity of the budget terms to individual processes has been explored in previous studies using the Frontier Research System for Global Change version of the University of California Irvine Chemical Transport Model (FRSGC/UCI CTM) in Wild (2007). One-at-a-time sensitivity runs were performed varying surface $NO_x$ emissions (30–60 TgN yr$^{-1}$), isoprene emissions (0–650 TgC yr$^{-1}$), lightning $NO_x$ emissions (0–7.5 TgN yr$^{-1}$), convective lifting, stratospheric influx and deposition processes (all ±50%), temperature (±5$^o$C) and humidity (±20%), and results are summarised in Fig. 1. This study highlighted the responses of a single model to particular processes, but the variations spanned relatively little of the parameter space defined by previous model studies, suggesting that substantial additional uncertainties were not accounted for here, including process interactions, neglected processes, and structural differences between models.

To explore the sensitivity of tropospheric budgets to uncertainty in several processes at once, we perform a global sensitivity analysis using Gaussian process emulation, following the approach of Lee et al. (2011). An emulator is a simple statistical model that reproduces the relationships between the inputs and outputs of a more complex model, in this case an atmospheric chemistry model. The much shorter run time of the emulator allows the model parameter uncertainty space to be explored fully through Monte Carlo approaches that would not be feasible with the complex atmospheric model. A Gaussian process is a multivariate normal distribution applied to a function, and we use this non-parametric approach to fit the model input-output relationships as it is well-tested, efficient and relatively easy to implement (O'Hagan, 2006; Lee et al., 2011; Ryan et al., 2018). This allows us to reproduce the nonlinear model response across a multidimensional parameter space based on a small ensemble of model training runs at points representing a combination of inputs that are optimally chosen to fill the space. We select eight key variables that influence global oxidant budgets substantially, and that span a range of model inputs (e.g., emissions), processes (e.g., deposition) and meteorological variables, see Table 2. These are based on our earlier one-at-a-time

**Table 2.** Variables and uncertainty ranges used in this study

| Variables | Range |
| --- | --- |
| Surface NOx emissions | 30–50 TgN yr$^{-1}$ |
| Lightning NO emissions | 2–8 TgN yr$^{-1}$ |
| Biogenic isoprene emissions | 200–800 TgC yr$^{-1}$ |
| Dry deposition rates | $\pm 60\%$ |
| Wet deposition rates | $\pm 60\%$ |
| Atmospheric humidity | $\pm 20\%$ |
| Cloud optical depth | $\times 0.33$–3.0 |
| Boundary layer mixing | $\times 0.10$–10.0 |

studies, and while they do not encompass all sources of uncertainty, which also include photochemical, transport and radiation processes, they are chosen to represent key uncertainties while ensuring that the study remains computationally tractable. We select surface emissions of $NO_x$ from natural and anthropogenic sources, the dominant precursor for $O_3$ in the troposphere; lightning emissions of NO, which are highly uncertain and have a disproportionately large impact on $O_3$ and OH due to the altitude of the source; and biogenic emissions of isoprene, which dominate global sources of volatile organic compounds. We include dry deposition, which is important for uptake of $O_3$ and other species at the surface, and wet deposition which is important for removal of soluble precursors. We vary the atmospheric humidity used by the model photochemistry, which plays an important role in $O_3$ chemistry and OH formation, but leave it untouched for other processes to avoid perturbing model dynamical processes. We vary cloud optical depth, an uncertain variable which has a major influence on photolysis rates in the lower troposphere. Finally, we vary turbulent mixing in the planetary boundary layer (PBL), which has an important role in lifting and dispersing surface oxidants, but which remains poorly constrained.

For each variable, we define a range that encompasses the maximum and minimum likely values that is loosely based on published studies from the literature, and these are presented in Table 2. We assume uncertainty ranges of $\pm 25\%$ for surface $NO_x$, representing 30–50 TgN yr$^{-1}$, $\pm 60\%$ for lightning NO (Schumann and Huntrieser, 2007) and $\pm 60\%$ for isoprene emissions (Ashworth et al., 2010). For dry and wet deposition, we assume an uncertainty in removal rates of $\pm 60\%$ that is applied to all species considered. We assume an uncertainty of $\pm 20\%$ for atmospheric water vapour, reflecting the variation across models contributing to the ACCMIP intercomparison (Lamarque et al., 2013), and this is applied in the model photochemistry scheme only. We assume an uncertainty of a factor of three in cloud optical depth based on Klein et al. (2013) and apply this for photolysis calculations only. Boundary layer mixing is perturbed by scaling the effective vertical diffusion coefficient through the depth of the boundary layer so that turbulent mixing of tracers between model layers varies from negligible to almost complete every model time step.

Following Lee et al. (2011), we use maximin Latin hypercube sampling to optimally select 80 points from across the eight-dimensional parameter space. Each point represents a combination of values chosen from the range for each variable, and

specifies the values to use for a full model simulation. An additional 24 points are selected to provide an independent test of the validity of the emulators that are built. This defines a set of 104 model simulations to perform. For this study, we use three independent global chemistry-transport models: the FRSGC/UCI CTM (Wild, 2007), the Goddard Institute for Space Studies Global Climate Model, GISS GCM (Shindell et al., 2013), and the Community Atmosphere Model with Chemistry, CAM-Chem (Lamarque et al., 2012). The models differ in their sources of meteorology, but are run for a full year (following 6–12 months spin-up) under conditions that are broadly consistent with 2001 meteorology, a year without strong climate phenomena such as El Niño. Offline meteorological fields for 2001 from the European Centre for Medium-Range Weather Forecasts Integrated Forecast System (ECMWF IFS) were used for the FRSGC/UCI CTM. The GISS GCM used observed sea-surface temperatures and was nudged to National Centers for Environmental Prediction (NCEP) reanalysis fields (Kalnay et al., 1996), while CAM-Chem was run in GCM mode following the Chemistry Climate Model Initiative (CCMI) REF-C1 protocol (Eyring et al., 2013). In each model we constrain methane to a fixed mixing ratio of 1760 ppb suitable for 2001 conditions. Natural and anthropogenic emissions differ somewhat across the models, reflecting different assumptions and online generation of natural emissions, but we scale the magnitude of global annual emissions to $40\,\mathrm{TgN\,yr^{-1}}$ for surface $NO_x$, $5\,\mathrm{TgN\,yr^{-1}}$ for lightning NO and $500\,\mathrm{TgC\,yr^{-1}}$ for isoprene in the control run, accepting that differences in emission distributions represent a source of structural uncertainty. Other variables are scaled according to the factors shown in Table 2 without further standardization between models.

Emulators are then built for each model for each output of interest using the methods described in Lee et al. (2011) and Ryan et al. (2018). We focus here on global annual mean tropospheric $O_3$ burden and $CH_4$ chemical lifetime for simplicity. The emulators are tested through use of the additional 24 validation simulations to evaluate their performance. For the outputs considered here, the model response surfaces are relatively smooth, reflecting the stable behaviour of the global $O_3$ burden and $CH_4$ lifetime, and the emulators fit the validation runs very closely with a correlation coefficient $r > 0.99$ (see Ryan et al., 2018). The emulators reproduce the response of the full model within the variable ranges defined, and can be used in place of the model for intensive analysis such as uncertainty propagation through the use of Monte-Carlo approaches that would not be computationally feasible with the full model. This allows us to define formal error bars for the response of each model, and to carry out global sensitivity analysis by determining the contribution of each variable to the overall variance in modelled $O_3$ burden and $CH_4$ lifetime.

## 3 Model responses and contributions to variance

We first use the emulators built for each model to propagate the uncertainty in the selected variables to uncertainty in $O_3$ burden and $CH_4$ lifetime. We use a Monte Carlo approach to randomly select 10,000 points from across the response space for each model, sampling uniformly across the full input range of each variable, and use this to generate the probability distribution for each model. Figure 2 shows the distribution in global $O_3$ burden and $CH_4$ lifetime from each model. The behaviour of the models is similar, with a normalised standard deviation of 7–8% for $O_3$ burden and 7–9% for $CH_4$ lifetime, and the distributions are slightly skewed, reflecting the nonlinear response of these budget terms to the governing processes. The $1\sigma$

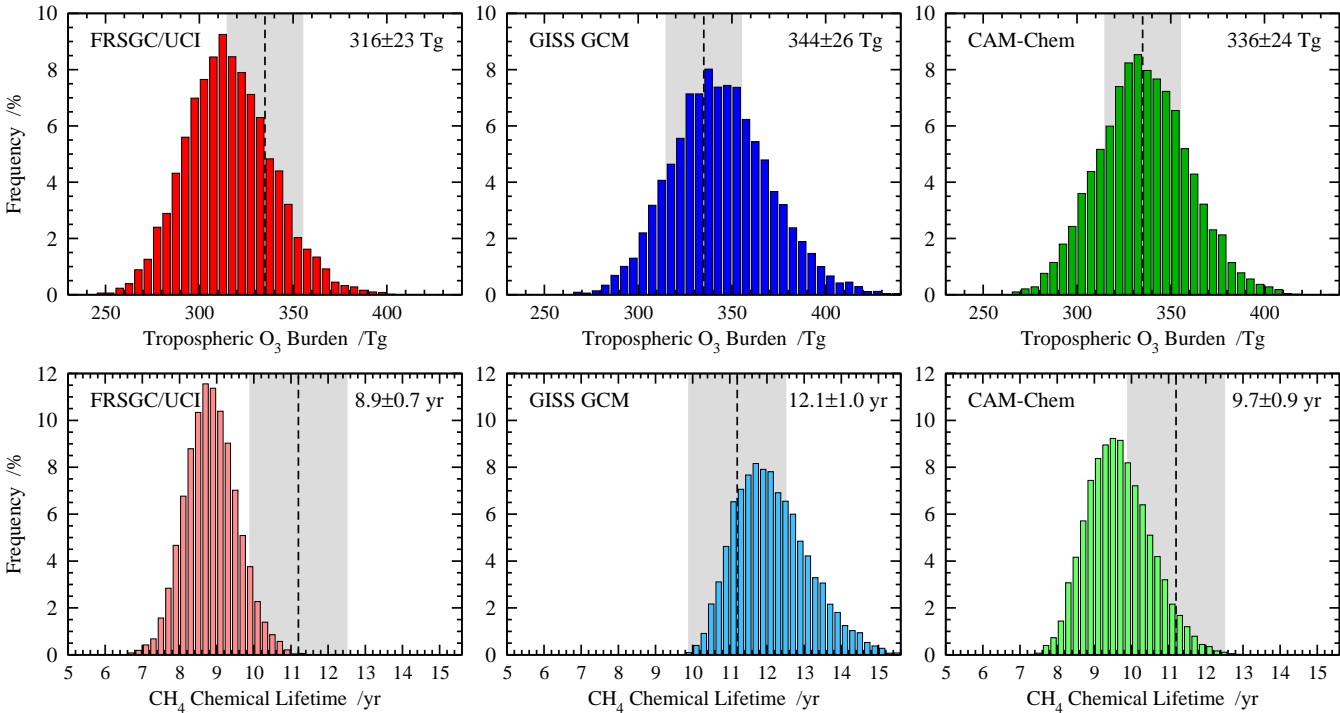

**Figure 2.** Probability distributions for the global annual mean tropospheric $O_3$ burden (top row) and tropospheric chemical lifetime of $CH_4$ (bottom row) for each model. The mean and standard deviation over 10,000 realizations are shown on the upper right of each panel, and observation-based estimates of $O_3$ burden and $CH_4$ lifetime are shown shaded.

uncertainty in each budget term is comparable in magnitude to that seen between different models in the ACCMIP model intercomparison (see Table 1); while this may be fortuitous, it demonstrates that process uncertainty contributes substantially to model diversity.

For each model, the mean $O_3$ burden lies within the observational uncertainty range, along with 44–60% of the distribution. A substantial proportion of each distribution lies outside the observational range, suggesting that the uncertainty ranges adopted for some of the variables were larger than needed, or that a normal distribution of uncertainty could have been assumed across each range in place of a uniform distribution. For mean $CH_4$ lifetime, agreement with observations is less good, with the GISS GCM and CAM-Chem lying at opposite boundaries of the observed range and the FRSGC/UCI CTM lying outside it.

For the GISS GCM, 63% of the distribution lies inside the observed range, while for the FRSGC/UCI CTM it is only 10%. The discrepancies between the modelled and observed estimates suggest that uncertainty in chemistry and transport processes, which have not been considered here, may play a substantial role in governing the $CH_4$ lifetime.

The sensitivity to each variable is determined by variance decomposition, which quantifies the contribution of each variable to the variance in the model output, and is shown in Fig. 3. This is performed through calculation of the sensitivity indices using

the Sobol approach (e.g., Saltelli, 2002), and the mathematical foundation for this is described in Ryan et al. (2018). We neglect

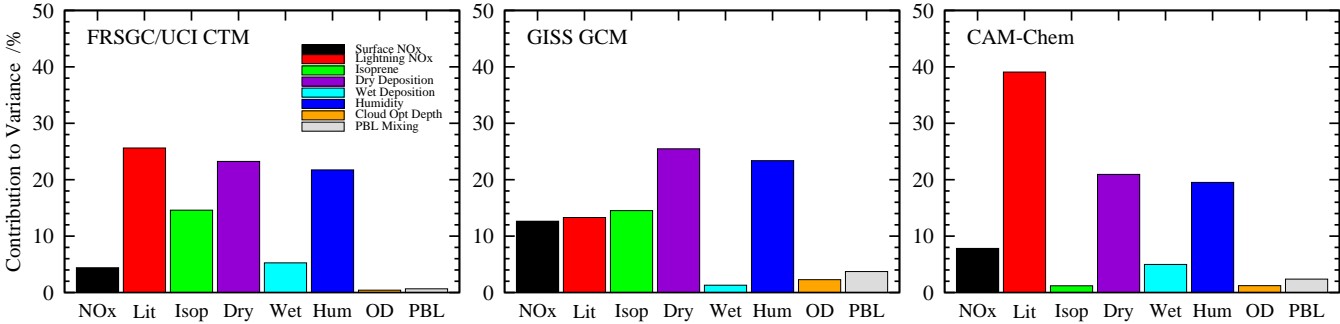

**Figure 3.** Contributions of each variable to the total variance in the simulated tropospheric $O_3$ burden in each model.

the contribution of interactions between variables, which can be identified through this approach but which remain below 4% of the variance for the model responses examined here. For the global $O_3$ burden, the models show relatively similar sensitivities to atmospheric humidity which contributes 20–23% of the variance in all three models, and to dry deposition processes which contribute 21–25%, see Fig. 3. However, there are substantial differences in sensitivities to lightning NO, which varies from 165     13% in the GISS GCM to 40% in CAM-Chem, and to isoprene emissions, which are 14% in FRSGC/UCI CTM and GISS GCM but only 1% in CAM-Chem. The consistent sensitivities to humidity and dry deposition are expected, given the important roles that these play as sinks of $O_3$ in the troposphere. A strong sensitivity to lightning NO is also expected given the greater chemical $O_3$ production efficiency of $NO_x$ in the mid- and upper troposphere, but the differing sensitivities between models likely reflect both differences in chemical environment and in lightning source distribution. Similarly, differences in sensitivity 170     to isoprene are likely to reflect differences in the complexity of the photochemical schemes in the models and in the resulting chemical environment in the tropical boundary layer.

For the tropospheric $CH_4$ lifetime, the models show notably different sensitivities, with humidity contributing about 20% of the variance for the FRSGC/UCI CTM and CAM-Chem, but less than 3% for the GISS GCM, see Fig. 4. There is broad consistency between the FRSGC/UCI CTM and CAM-Chem, where uncertainty in lightning NO is the largest contributor 175     and emissions of isoprene and surface $NO_x$ are about 30% and 50% less, respectively, but in the GISS GCM the strongest sensitivity is to surface $NO_x$ emissions. It is clear that the factors governing tropospheric OH are substantially different in the models, highlighting differences in chemical environment and transport patterns that affect the location and magnitude of $CH_4$ oxidation. Sensitivity to humidity suggests that primary sources of OH through photolysis of $O_3$ and subsequent reaction of $O^1D$ with water vapour are important. Sensitivity to $NO_x$ emissions reflects the importance of secondary sources of OH 180     through oxidation of NO, and sensitivity to isoprene highlights the importance of VOC as a source and sink of OH and as a mechanism for locking up and transporting $NO_x$. Interestingly, the GISS GCM shows substantial sensitivity to boundary layer mixing, highlighting the importance of transport of fresh emissions from the surface for secondary OH formation. The FRSGC/UCI CTM shows some sensitivity to wet deposition, suggesting that scavenging of nitric acid has a direct impact on OH through its influence on the abundance of $NO_x$.

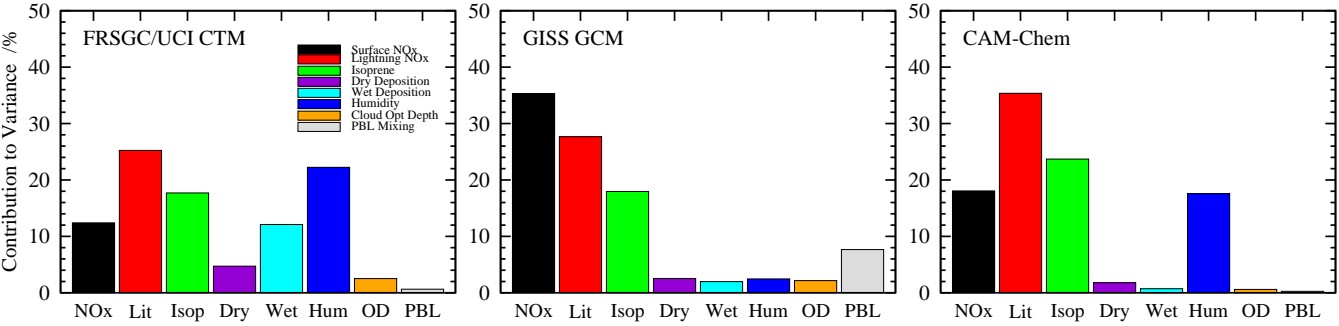

**Figure 4.** Contributions of each variable to the total variance in the simulated annual mean $CH_4$ chemical lifetime in each model.

These differences have important implications for assessment of future composition change. Future scenarios projecting increased emissions of greenhouse gases and reduced emissions of $O_3$ precursors (e.g., RCPs 4.5, 6.0 and 8.5) are likely to lead to increased future humidity and reduced surface $NO_x$. The FRSGC/UCI CTM and CAM-Chem would be expected to show a reduction in $CH_4$ lifetime due to greater OH concentrations associated with higher water vapour, while the GISS GCM would show an increase in $CH_4$ lifetime due to lower secondary production of OH associated with reduced surface $NO_x$ emissions. Analysis of future changes in $CH_4$ lifetime for models contributing to the ACCMIP intercomparison suggests that this is indeed the case, with the GISS GCM one of three models showing increased lifetime by 2100 for the RCP6.0 pathway, and four models showing decreased lifetime (Voulgarakis et al., 2013). An understanding of the causes of this differing sensitivity is thus important for explaining the different model responses.

## 4 Investigating model differences

The sensitivity of modelled $O_3$ burden and $CH_4$ lifetime to two key variables, humidity and surface $NO_x$ emissions, is shown for the FRSGC/UCI CTM and GISS GCM in Fig. 5. These response surfaces are generated using the emulator for each model assuming that the other six variables are unchanged. While the $O_3$ burden is slightly higher in the GISS GCM than the FRSGC/UCI CTM, 342 vs 314 Tg in the model control runs, the gradients across the response surfaces are similar in the models. The highest $O_3$ burdens occur at high $NO_x$ emissions and low humidity, reflecting greater production and reduced loss, respectively. The relative changes in $O_3$ burden with $NO_x$ emissions and humidity are very similar across all three models, as shown in Fig. 6. The responses for $CH_4$ lifetime show notably different behaviour, with greater sensitivity to $NO_x$ and much less sensitivity to humidity in the GISS GCM compared to the other models. At high humidities the $CH_4$ lifetime appears almost insensitive to humidity, suggesting either little additional formation of OH or a matching OH sink in this model. In contrast, the other models show a very similar degree of sensitivity to humidity in both $O_3$ burden and $CH_4$ lifetime that ranges from +7% to -5% across the humidity range considered here, see Fig. 6. This suggests a much stronger coupling between $O_3$ and OH formation, and highlights the greater importance of the primary OH source in these models.

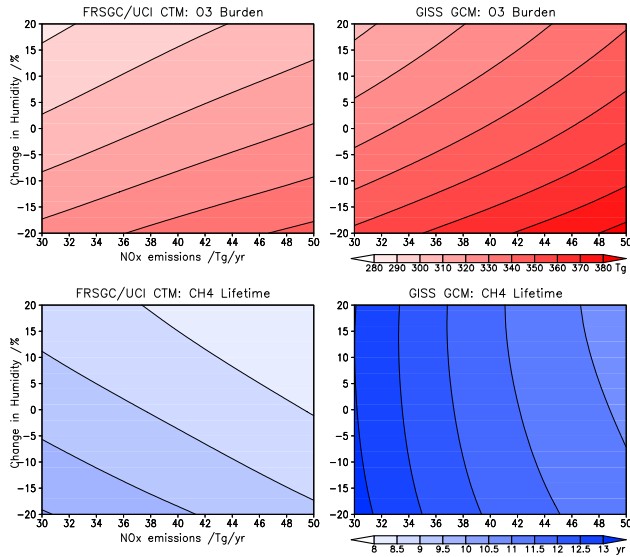

**Figure 5.** Sensitivity of tropospheric $O_3$ burden and $CH_4$ chemical lifetime to changes in surface $NO_x$ emissions and humidity in the FRSGC/UCI CTM and GISS GCM.

The response surfaces shown here allow us to estimate the impact of changes in future humidity and surface $NO_x$ emissions in the absence of other changes. A reduction in $NO_x$ emissions from 40 to $30 \, \text{TgN} \, \text{yr}^{-1}$ and increase in humidity of 15%, corresponding loosely to the changes between 2000 and 2050 expected along the RCP8.5 pathway (van Vuuren et al., 2011),
would lead to an increase in $CH_4$ lifetime of 1.3 yr in the GISS GCM (from 11.7 to 13.0 yr), an increase of 0.2 yr in CAM-Chem, and no change in the FRSGC/UCI CTM. While this neglects the influence of other emission and climate changes, particularly the increase in $CH_4$ concentrations which would extend the lifetime in all models, it demonstrates the very different sensitivities anticipated for different models under future climate scenarios.

To help identify the cause of the differing model responses, we show the contribution of key variables to the variance in the
annual mean tropospheric column $CH_4$ chemical loss rate at each model grid point in Fig. 7. This shows how the contribution of the different processes governing $CH_4$ removal varies geographically and reveals further differences between the models. For the FRSGC/UCI CTM and CAM-Chem, humidity makes an important contribution to the variance in tropical regions and at mid-latitudes, and makes a smaller contribution at the equator, where the greatest contribution is from lightning NO in all three models. Humidity makes very little contribution to the variance in the GISS GCM, and this principally occurs downwind
of major anthropogenic emission regions. The underlying humidities in the models are relatively similar (see distributions presented in the supplement), and the annual mean global atmospheric water burden is also similar, only 4% less in the GISS GCM than in the FRSGC/UCI CTM. Given the similar humidities and similar responses in $O_3$ burden, this suggests that there are significant differences in chemical processes specific to OH. Despite the larger relative importance of surface $NO_x$ emissions in the GISS GCM, the absolute contribution to the variance in the three models is similar. Surface $NO_x$ emissions
have a widespread impact, contributing substantially to $CH_4$ removal over remote ocean regions. The effect of $NO_x$ on OH

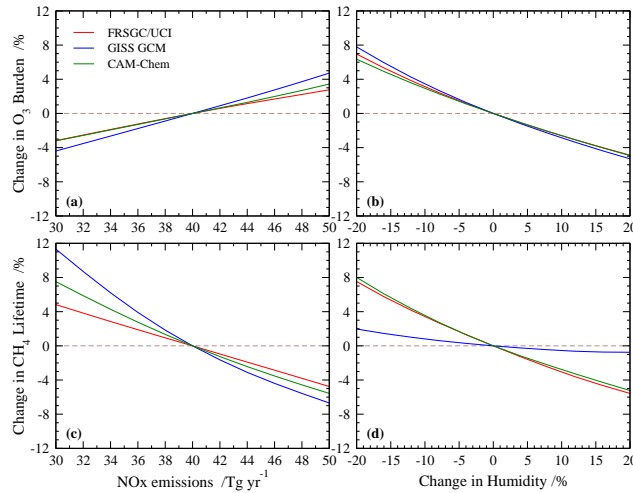

**Figure 6.** Relative changes in tropospheric $O_3$ burden **(a, b)** and $CH_4$ chemical lifetime **(c, d)** to changes in surface $NO_x$ emissions and humidity alone in each model.

in these locations suggests that substantial nitrogen is transported to these regions in the form of reservoir species such as peroxyacetyl nitrate (PAN), and this is supported by the patterns of transport seen in the isoprene contribution. The greatest effect of isoprene emissions is localised in the tropical continental source regions due to the relatively short lifetime of isoprene and its oxidation products, but there are substantial contributions downwind over the oceans, particularly in CAM-Chem and the GISS GCM. Mid-tropospheric PAN concentrations are much greater in the GISS GCM, and comparison of tropospheric $NO_2$ columns suggest that there are higher levels of $NO_x$ over oceanic regions in this model (see supplementary material). It is therefore likely that differing treatments of $NO_y$ chemistry are one cause of the different model sensitivities. However, a more detailed exploration of the sensitivity to photochemical processes would be needed to confirm this. Tropospheric OH is dependent on the total ozone column in the tropics through its effect on photolysis rates, and this may play a role in model differences, although we note that mean tropical ozone column in the present models is very similar at 258–265 DU (see Table S2). Underlying differences in meteorological fields governing vertical transport processes such as convection are also likely to be important in this region. Our analysis provides a valuable guide to locations where model responses are likely to differ most, such as in tropical oceanic regions, and further investigation of OH sensitivity in these regions should bring improvements in our understanding of atmospheric processes and in their representation in current global-scale models.

## 5   Conclusions

We have demonstrated the value of Gaussian process emulation in performing global sensitivity analysis of computationally-intensive global atmospheric chemistry transport models, and in applying this across a number of models to investigate model diversity. The approach provides a simple way of exploring the sensitivity of key terms in the tropospheric oxidant budget to

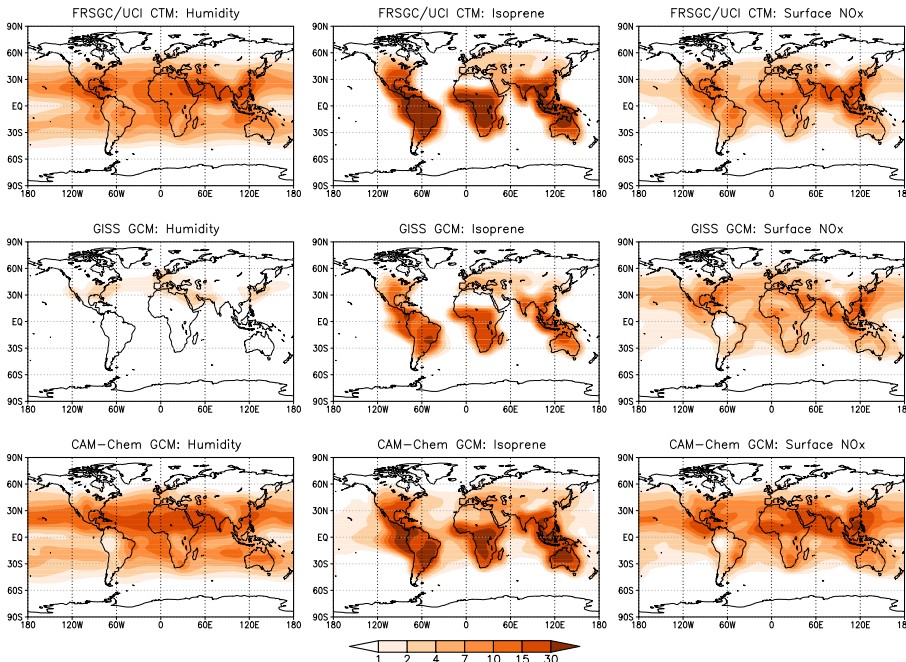

**Figure 7.** Contributions to the total variance in the annual tropospheric column $CH_4$ chemical loss rate (in $\mathrm{mg\,m^{-2}\,yr^{-1}}$) in each model from humidity, isoprene emissions and surface $NO_x$ emissions. Fractional contributions (sensitivity indices) are presented in Fig. S3 in the supplement.

governing processes and inputs, and we show that it can provide substantial new insight into the differing responses of models
under different emission and climate scenarios.

Our study has highlighted the large sensitivity of the tropospheric $O_3$ burden to atmospheric water vapour, suggesting that this variable should be diagnosed or perhaps constrained in future model intercomparisons to permit clearer characterization of differences in model chemistry. We also find a strong sensitivity to precursor emissions and to dry deposition processes, as expected. More surprisingly, we find that the drivers of variability in global OH can be very different between models, and this
may contribute to the large diversity in modelled tropospheric $CH_4$ lifetimes seen in recent model intercomparisons. Given the importance of atmospheric oxidising capacity for both air quality and climate change, this difference in OH behaviour is a major cause for concern and is a clear priority for further investigation.

While we have shown the value of emulation approaches for exploring model behaviour much more thoroughly than through simple one-at-a-time sensitivity studies, this study has been largely exploratory in nature, investigating the effects of a very
limited number of variables. A more detailed global uncertainty analysis is required that considers a wider range of model processes and inputs and incorporates a more rigorous assessment of uncertainty in each variable. Application of observation-based constraints is then needed to restrict the size of the response space to calibrate the models and identify specific processes in need of refinement. Applying this approach across different models accommodates the structural uncertainties in model

formulation, permitting a more robust assessment of process understanding. This would provide a strong evaluation framework for improving understanding of the physical and chemical processes driving atmospheric composition change and its effects on air quality and climate.

*Data availability.* The monthly mean output from each model for the ensemble of runs performed in this study will be made available from the CEDA data archive, and can be accessed by request to the corresponding author.

*Author contributions.* OW, LL, FO and AV designed the study. OW, AV and JL ran model simulations, and ER and LL performed the analysis. OW prepared the manuscript with contributions from all co-authors.

*Competing interests.* The authors declare that they have no conflict of interest.

*Acknowledgements.* This work was supported in part by the UK Natural Environment Research Council [grant number NE/N003411/1]. AV thanks the NASA High-End Computing (HEC) Program through the NASA Center for Climate Simulation (NCCS) at Goddard Space Flight Center for providing computational resources to perform GISS model simulations, and Greg Faluvegi from Columbia University/NASA GISS for advice on setting up the GISS model experiments. CAM-Chem is a specific configuration of CESM, which is supported primarily by the National Science Foundation. Computing and data storage resources, including the Cheyenne supercomputer (doi:10.5065/D6RX99HX), were provided by the Computational and Information Systems Laboratory (CISL) at NCAR, which is a major facility sponsored by the National Science Foundation under Cooperative Agreement No. 1852977.

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
