# Peer review of "Global sensitivity analysis of chemistry-climate model budgets of tropospheric ozone and OH: Exploring model diversity"

_Atmospheric Chemistry and Physics, 2019_

## Referee Comment (RC1) · Anonymous Referee #1 · 8 Oct 2019

The manuscript presents an analysis of the response of tropospheric ozone and OH to a number of different factors across three different global chemistry models. A number (order 80) of sensitivity experiments varying factors such as specified emissions, deposition rates and atmospheric humidity are specified and Gaussian process emulation is used to extend the model response across the full parameter space. Sensitivity of the model tropospheric ozone burden is found to be dominated by humidity, while two of the three models also show the prescribed variations in humidity also dominate the response of OH. Interestingly, the third model shows almost no response of OH to the imposed variations in humidity and, correspondingly, has a larger variation deriving from other factors such as surface emissions of NOx and isoprene, and lightning NOx.

The manuscript is very well written and the results and conclusions are clearly laid out. The findings of the importance of the effect of water vapour for tropospheric OH and ozone and that one of the three models shows a radically different sensitivity for OH are certainly of great interest to the global chemistry modelling community as the causes of the diversity across models is a long-standing problem. The only significant comment I would have is that given the somewhat arbitrary specifications for some of the ranges over which the different factors are varied, it is very difficult to put these results into context. It is shown that varying water vapour by $\pm$ 50% is the dominant factor affecting ozone and OH, but how does the range of $\pm$ 50% compare to the actual variability across models? While emissions, including lightning NOx, have well defined and regularly discussed ranges across models and other processes such as boundary layer mixing are very poorly constrained, a range for water vapour across models should be more easily quantified and given the importance of the process found here it should be more clearly justified. From a quick look at Figure 7 of Lamarque et al. (Geosci. Model Dev., 6, 179–206, 2013) I think a variation of $\pm$ 50% might be too large.

My other minor comments are given below.

Lines 32: While I understand the need for brevity here, I do find the statement that 'changes in surface O3 since the preindustrial era are systematically underestimated (Stevenson et al., 2013)' to be a bit of an over-simplification of the situation. Recent work under TOAR (Tarasick et al., under review) and work with oxygen isotopes (Yeung et al, Nature, doi:10.1038/s41586-019-1277-1, 2019) have found new reasons why the increase in tropospheric ozone may not be as large as shown by the early surface measurements. I am not suggesting an exhaustive discussion here, but just some acknowledgement that there is uncertainty.

Line 109 – 110: How do you approach increasing wet deposition rates for a species such as HNO3?

Lines 176 – 177: In both Figures 5 and 6 the very different response of OH in GISS to

humidity is highlighted, but because humidity is only displayed as a percentage change relative to the baseline it is not possible to ascertain whether the behaviour of GISS is due to some fundamentally different response of the chemical scheme to water vapour or whether the response is due to the models being in a different part of the parameter space. Given the importance of water vapour, is it possible to provide some absolute comparison of water vapour across the three models? I'll also note that while the full range of NOx emissions are plotted for these two figures, water vapour is only plotted as $\pm$ 20% while the range of the sensitivity experiments is $\pm$ 50%.

Lines 198 – 202: In discussing Figure 7 and the different response of GISS, there is speculation that GISS may have a very different treatment of NOy chemistry because of the more widespread sensitivity to surface NOx and isoprene emission, particularly in the tropics. The quantity plotted here is the percent variance of CH4 loss that can be assigned to each of the different factors. Since water vapour in the GISS model explains almost none of the variance in the tropics, doesn't the variance have to be assigned somewhere else. Is, in some absolute measure, methane loss in the GISS model more sensitive to NOx and isoprene emissions than the other two models or does it only assign more of the response to NOx and isoprene emissions because of the negligible contribution from water vapour?

---

## Referee Comment (RC2) · Anonymous Referee #2 · 16 Oct 2019

This is an interesting paper where "identifying and quantifying the main causes of diversity in current models" is attempted through Gaussian process elimination. Three models are analyzed for tropospheric methane lifetime and ozone burden: the FRSGC/UCI CTM, the Goddard Institute for Space Studies Global Climate Model, GISS GCM, and the Community Atmosphere Model with Chemistry, CAMChem. The GISS GCM shows a rather different sensitivity to methane oxidation than the other two models, although the reasons for this are never clearly articulated. The sensitivity in the different models can explain some of their different responses to global change over the next century.

Overall, while this paper demonstrates a unique and potentially powerful analysis tech-

nique. However, I was not altogether satisfied with some of the analysis or in the end with the scientific depth of the paper.

Major Comments:

1. How were the key variables selected for the Gaussian process elimination? The selection of variables should be justified. For example, Holmes et al. (2012) (No, I am not an author on this paper) found that OH is most sensitive to the ozone column (at least when examining interannual variability). This was not included in the present study. This choice needs to be justified.

2. The selected range of the sensitivity variables is very important for the paper particularly when comparing the sensitivity of one variable against another. The paper states the ranges are loosely based on studies in the literature, but do not give the studies. To me, at least it is not believable that cloud optical depth varies could vary by 100, or boundary layer mixing by a factor of 10,000. Of more concern is that humidity varies globally by 50%. This is in fact very large and colors the results and conclusions throughout the paper. In contrast, Holmes et al (2012) gives a variation of 3% in humidity. While the variability might be calculated differently in the two studies, a value of 50% seems huge. Would the range in variability be better quantified by comparing across the models? I also do not believe it is reasonable to take the variability from the smallest and largest values in the literature as this does not likely capture the likely error in model simulations. In any case, the sensitivity and ranges in variability need to be better quantified as this impacts much of the interpretation in the paper. An arbitrary specification of the range in these parameters would seem to imply that the resulting comparison of the different sensitivities is also arbitrary.

3. Comparing the sensitivities across models only makes sense if the model forcing is similar. (For example, the sensitivity of any one model might be very different when comparing between present day conditions and pre-industrial conditions). The authors need to show that the tropospheric forcing in the three models is roughly the same

(CH4, NOx emissions, tropical ozone column etc). In particular I am concerned about the tropical ozone column and perhaps more importantly the tropical photolysis rates. Might this explain the large difference in the methane oxidation rates between the different models? At any rate quantifying the mean differences (at least in a supplement) seems important in better understanding the results.

4. The paper does not really pinpoint some of the basic causes of the discrepancies between the models, in particular the different lifetimes of methane. As stated in the previous comment some of the mean fields in the models should be given, for example the global burden of CH4, atmospheric water vapor (perhaps with latitude?), etc. While the sensitivity tests between the different models is revealing evaluating the mean difference between the GISS-GCM and the other models may also be revealing. In a number of locations the authors hypothesize that the difference is due to differences in humidity (lines 177-178, Pages 8 and Page9) or due to differences in the formation/decomposition of PAN. Why don't the authors check? I'm not suggesting a lengthy analysis, but an inspection and comparison of some of the mean fields should be sufficient to check some of these hypothesis and possibly reveal some of the key model differences.

Minor Comments

1. Please state in the caption to Figure 1 that these results are over different years with different emissions and meteorologies. Also please reference Table 1 in the Figure caption where the references to the points can be found. Finally, please state what the dots refer to in the right hand panel.

2. The methodology behind the Gaussian process elimination should be explained in more detail in methodology section. While I don't expect the authors to go into detail, neither should it be necessary to reference the referenced papers to understand this analysis.

3. Does the sensitivity to boundary layer mixing simply involve changing the vertical

diffusion coefficient over the boundary layer depth, or something else?

4. The paper states: "The models differ in their sources of meteorology", but certainly the sources of meteorology are important. Are the runs using specified dynamics or GCM generated meteorology? This might be quite important to how well the relative humidity is specified. In addition, using a single year might also introduce significant variability between the models. Please say something more specific about the meteorology used as well as comment on the possible importance of the interannual variability. My guess is the latter might introduce significant variability between the models.

5. Page 5, line 15. "but we scale". It is not altogether clear how this scaling is used. Please specify. Given the non-linear response to NOx, does a linear scaling make sense?

6. Page 7, line 139-141 "discrepancies highlight that uncertainty in chemistry and transport processes not considered here may play a substantial role in governing the CH4 lifetime". I'm not sure I understand this conclusion. It seems it might simply mean, for example, that the variability in water vapor is considerably over-estimated.

7. Page 7, line 149 "it is notable that humidity has not been prescribed in previous model intercomparison studies". Possibly, but again the large response in water vapor is dependent on the large variation in water vapor.

8. Page 8. The explanation in the future responses of the different models is nice and given in several places in the paper. Personally, I think this might belong better in the conclusion section.

9. Page 9, line 180 "suggesting a saturation in OH formation in this model". How does a saturation occur?

10. More generally, I am curious about the conceptual difference between the sensitivity to forcing parameters (NOX emissions, stratospheric ozone column) and internal model parameters. To what extent does it make sense to distinguish between the sensitivity between these type of parameters?

---

## Referee Comment (RC3) · Anonymous Referee #3 · 24 Oct 2019

In this paper, the authors explore the sensitivity of tropospheric ozone and methane lifetime to different factors in three global chemistry-transport models using an emulation process with the goal of identifying the causes of diversity in the model response to changing forcings and climate. This study is an important contribution towards understanding the reasons for model diversity in the evolution of tropospheric ozone and methane lifetime. However, I found that the approach applied here is inadequate to truly understand the reasons for diversity in these non-linear quantities. My main concern (similar to the other two reviewers) is that the sensitivities calculated for each model would depend on the "control" simulation given the non-linear chemistry of ozone and methane. If the models differ in the forcings (meteorology and emissions) to begin

with then how do we know that the calculated sensitivity is not due to the initial state. Also, I found the description of the Gaussian emulation approach applied here rather limited to appreciate its usefulness for understanding the reasons for diversity in model response.

Below are some specific comments and suggestions to help strengthen the paper.

L23-25: Clarify if this is referring to controls for climate or air pollution. Controls on NOx emissions in the US (e.g., Clean Air Act) and Europe (e.g., LRTAP) have indeed brought down surface ozone.

L27: Also another ACCMIP paper (Naik et al., 2013) and CCMI models (Zhao et al., 2019 https://www.atmos-chem-phys-discuss.net/acp-2019-281/)

L32: There are large uncertainties in PI estimates of surface ozone as discussed by Tarasick et al. (2019) https://www.elementascience.org/articles/10.1525/elementa.376/

L58: Observational estimates of global ozone are now available from satellites (Gaudel et al. 2019 https://www.elementascience.org/articles/10.1525/elementa.291/). How do the model estimates discussed here compare with satellite estimates?

L100-104: It would help to know how different the base state is in the models. What is the ozone burden, prescribed methane concentration, methane lifetime, surface and lightning NOx emissions, biogenic emissions, wet and dry deposition rates for all species, atmospheric humidity, cloud optical depth, and boundary layer height in the base simulation for all the three models?

L114-115: How are the emulators built for a non-linear system such the O3-NOx-CH4 chemistry? Some description is needed to make the design of emulators transparent for the purpose of this figure.

L142-143: It would be helpful to provide an equation to explain how sensitivity for each variable is determined. As it stands, the process appears too opaque to me.

[Figure]

L151: How different is humidity across the three models for the base run? Is it possible that the three models show large sensitivity of ozone to humidity because such a large ($\pm$ 50%) perturbation is used? How do the sensitivities for ozone calculated here compare with those calculated by Revell et al. (2018) https://www.atmos-chem-phys.net/18/16155/2018/acp-18-16155-2018.pdf?

L167-169: "and four models showing decreased lifetime" - is the implication here that these four models may have greater sensitivity to humidity and therefore show declining methane lifetime? If so, how do we know that these models are like CAM-chem and FRSGC/UCI CTM in their sensitivities?

L192-194: The chemical loss of methane also depends on the concentration of methane in the models. Are they the same across the models?

L198: How different are the model chemical mechanisms implemented in the models? I would imagine the differences in sensitivities due to NOx are due to the implemented chemical mechanisms.

---

## Author Comment (AC1) · 15 Jan 2020

**Response to reviewers on "Global sensitivity analysis of chemistry-climate model budgets of tropospheric ozone and OH: Exploring model diversity" by Wild et al.**

**Response to Reviewer 1:**

*The manuscript is very well written and the results and conclusions are clearly laid out. The findings of the importance of the effect of water vapour for tropospheric OH and ozone and that one of the three models shows a radically different sensitivity for OH are certainly of great interest to the global chemistry modelling community as the causes of the diversity across models is a long-standing problem.*

We thank you for your very positive comments here, and address specific concerns below.

*The only significant comment I would have is that given the somewhat arbitrary specifications for some of the ranges over which the different factors are varied, it is very difficult to put these results into context. It is shown that varying water vapour by ±50% is the dominant factor affecting ozone and OH, but how does the range of ±50% compare to the actual variability across models? While emissions, including lightning NOx, have well defined and regularly discussed ranges across models and other processes such as boundary layer mixing are very poorly constrained, a range for water vapour across models should be more easily quantified and given the importance of the process found here it should be more clearly justified. From a quick look at Figure 7 of Lamarque et al. (Geosci. Model Dev., 6, 179206, 2013) I think a variation of ±50% might be too large.*

The parameter ranges we are working with are intentionally large to capture the full range of uncertainty in each variable, and this provides us with response surfaces that fully encompass the likely uncertainty. However, we agree that realistic, consistently-defined uncertainty ranges are needed when comparing model sensitivity across different variables. These are more difficult to determine for model processes than for emissions as the reviewer notes. The likely uncertainty in water vapour across models is of the order of ±20% from the Lamarque et al. 2013 study (which corresponds to temperature biases of about 2.5 K) and we implicitly acknowledged this in our presentation of water vapour in Fig. 5, which the reviewer noted.

To address this issue, which the other reviewers also raise, we have repeated the sensitivity analysis over a more restricted and directly comparable range of variability for each parameter. We maintain the existing ranges for emissions, which are well grounded in the literature, reduce the deposition ranges from ±80% to ±60%, although these remain highly uncertain, and reduce the range for water vapour to ±20%. We reduce the range for cloud optical depth to a factor of 3 based on Klein et al. [2013], and restrict the range for boundary layer mixing to a factor of 10, which still encompasses both very weak and instantaneous mixing, both of which are simple options that have been adopted in previous model studies. The new sensitivities are now presented in the paper; while these differ from those over the full ranges originally presented, with greater importance for emissions at the expense of humidity, our key messages about the importance of water vapour and about the differing responses in OH across the models remain unchanged, see Figure 1 below.

[Figure]

**Fig 1: Contributions of each variable to the total variance in the simulated tropospheric O$_3$ burden (top row) and annual mean CH$_4$ chemical lifetime (bottom row) in each model over the revised sensitivity range.**

To avoid complicating the paper with two sets of parameter ranges, one over which the models were run, and one for the sensitivity analysis, we have adjusted the text to focus on the new ranges, but include the results from the full ranges in supplementary material. We highlight that this is an exploratory study demonstrating the value of the approaches adopted here, and that the contribution of each parameter to the overall sensitivity is dependent not only on the ranges selected but also on which parameters are considered. A much wider choice of parameters is needed for a more in-depth study with greater explanatory power, and this is noted in the paper.

*Lines 32: While I understand the need for brevity here, I do find the statement that changes in surface O$_3$ since the preindustrial era are systematically underestimated (Stevenson et al., 2013) to be a bit of an over-simplification of the situation. Recent work under TOAR (Tarasick et al., under review) and work with oxygen isotopes (Yeung et al, Nature, doi:10.1038/ s41586-019-1277-1, 2019) have found new reasons why the increase in tropospheric ozone may not be as large as shown by the early surface measurements. I am not suggesting an exhaustive discussion here, but just some acknowledgement that there is uncertainty.*

These new papers highlight the uncertainty in preindustrial ozone, and the reviewer is correct to point this out. We have toned down this statement to reflect this uncertainty, and now refer the reader to the Tarasick et al. paper which discusses this in detail. We add "although there is continuing uncertainty surrounding preindustrial levels (Tarrasick et al., 2019)."

*Line 109–110: How do you approach increasing wet deposition rates for a species such as HNO$_3$?*

Wet deposition rates are altered by scaling the removal rate for each soluble tracer affected by washout and rainout each model timestep. This is a crude but simple approach that can be implemented consistently across models. While it has the potential to create instability

where removal rates are greatly increased, the scalings used here are relatively small. A more in-depth exploration would involve varying cloud water content, rainfall and the solubility of different species independently. We have rephrased the text to make this clearer.

*Lines 176–177: In both Figures 5 and 6 the very different response of OH in GISS to humidity is highlighted, but because humidity is only displayed as a percentage change relative to the baseline it is not possible to ascertain whether the behaviour of GISS is due to some fundamentally different response of the chemical scheme to water vapour or whether the response is due to the models being in a different part of the parameter space. Given the importance of water vapour, is it possible to provide some absolute comparison of water vapour across the three models? Ill also note that while the full range of NOx emissions are plotted for these two figures, water vapour is only plotted as ±20% while the range of the sensitivity experiments is ±50%.*

The underlying humidities in the models are relatively similar, and the sensitivities for ozone (Fig 6b) are also similar, suggesting that the differences for OH arise from chemical processes rather than from humidity. However, we appreciate that different model base states may be one reason for differing sensitivities, and we have therefore included a comparison of key variables between the models in supplementary material, as suggested. This shows a high degree of similarity between the models, but points to differences in NOx and PAN that we discuss in the paper. The mean mass of water vapour in the GISS GCM runs ($1.21 \times 10^{16}$ kg) is about 4% less than than in the FRSGC/UCI CTM runs ($1.26 \times 10^{16}$ kg), but this is well within the uncertainty range we explore. We now include these details in the paper. As noted above, our restriction to a ±20% variation for water vapour tacitly acknowledges this more realistic assessment of the likely uncertainty in this variable, and we now use ±20% throughout the study.

*Lines 198–202: In discussing Figure 7 and the different response of GISS, there is speculation that GISS may have a very different treatment of NOy chemistry because of the more widespread sensitivity to surface NOx and isoprene emission, particularly in the tropics. The quantity plotted here is the percent variance of $CH_4$ loss that can be assigned to each of the different factors. Since water vapour in the GISS model explains almost none of the variance in the tropics, doesnt the variance have to be assigned somewhere else. Is, in some absolute measure, methane loss in the GISS model more sensitive to NOx and isoprene emissions than the other two models or does it only assign more of the response to NOx and isoprene emissions because of the negligible contribution from water vapour?*

This is a valuable point, and the reviewer is correct to note that the absolute variance is also of interest. We have recalculated the contributions using the new parameter ranges, and now present the absolute variances in Figure 7, placing the relative contributions (more formally, the sensitivity indices) in the supplementary material. It is much clearer from this analysis that the sensitivity to NOx and isoprene emissions in the GISS GCM is comparable to that in the other models, but that the sensitivity to humidity is substantially less. We have adjusted the discussion in the paper to reflect this point, and show the revised figure as Figure 2 below.

[Figure]

**Fig 2: Contributions of each variable to the absolute variance in the simulated tropospheric O$_3$ burden (top row) and annual mean CH$_4$ chemical lifetime (bottom row) in each model over the revised sensitivity range.**

**Response to Reviewer 2:**

*1. How were the key variables selected for the Gaussian process elimination? The selection of variables should be justified. For example, Holmes et al. (2012) (No, I am not an author on this paper) found that OH is most sensitive to the ozone column (at least when examining interannual variability). This was not included in the present study. This choice needs to be justified.*

This is an exploratory study and we wanted to investigate a range of different inputs and model processes, but were limited to eight variables for computational expediency and to permit application across unrelated models. The variables we selected were based on one-at-a-time sensitivity studies presented in Wild et al. [2007], as explained in paragraph 4 of Section 2 (lines 83–95), where the impacts on ozone and OH were demonstrated to be substantial. There are many other important variables, including the ozone column, that we were not able to consider. Now that we have established the value of the approach in comparing sensitivies across models, we will explore a more complete range of variables over a larger parameter space. We highlight the necessarily restricted scope of this study in the text, and have now revised it to acknowledge the importance of other variables more clearly.

*2. The selected range of the sensitivity variables is very important for the paper particularly when comparing the sensitivity of one variable against another. The paper states the ranges are loosely based on studies in the literature, but do not give the studies. To me, at least it is*

*not believable that cloud optical depth varies could vary by 100, or boundary layer mixing by a factor of 10,000. Of more concern is that humidity varies globally by 50%. This is in fact very large and colors the results and conclusions throughout the paper. In contrast, Holmes et al (2012) gives a variation of 3% in humidity. While the variability might be calculated differently in the two studies, a value of 50% seems huge. Would the range in variability be better quantified by comparing across the models? I also do not believe it is reasonable to take the variability from the smallest and largest values in the literature as this does not likely capture the likely error in model simulations. In any case, the sensitivity and ranges in variability need to be better quantified as this impacts much of the interpretation in the paper. An arbitrary specification of the range in these parameters would seem to imply that the resulting comparison of the different sensitivities is also arbitrary.*

We selected large ranges to emulate to ensure that the resulting model response surfaces capture the full range of uncertainty possible for each variable. However, we agree that realistic, consistently-defined uncertainty ranges are needed when comparing model sensitivity across different variables, as the reviewer notes. In particular, the range for humidity is indeed large, and climate model intercomparisons suggest an uncertainty in water vapour of the order of $\pm 20\%$ (see Lamarque et al., 2013). To address the reviewer's concerns, we have repeated the sensitivity analysis using restricted ranges for the variables that are more closely aligned with previous assessments in the literature, and we cite the studies concerned. We maintain the existing ranges for emissions, which are well grounded in the literature, reduce the deposition ranges from $\pm 80\%$ to $\pm 60\%$, and reduce the range for water vapour to $\pm 20\%$. We reduce the range for cloud optical depth to a factor of 3 based on Klein et al. [2013], and restrict the range for boundary layer mixing unchanged to a factor of 10, which still encompasses both very weak and instantaneous mixing, both of which are simple options that have been adopted in previous model studies. The new sensitivities reflect a greater importance for emissions at the expense of humidity, as expected, but our key messages about the importance of water vapour and about the differing responses in OH across the models remain unchanged.

*3. Comparing the sensitivities across models only makes sense if the model forcing is similar. (For example, the sensitivity of any one model might be very different when comparing between present day conditions and pre-industrial conditions). The authors need to show that the tropospheric forcing in the three models is roughly the same ($CH_4$, NOx emissions, tropical ozone column etc). In particular I am concerned about the tropical ozone column and perhaps more importantly the tropical photolysis rates. Might this explain the large difference in the methane oxidation rates between the different models? At any rate quantifying the mean differences (at least in a supplement) seems important in better understanding the results.*

We standardised the model simulations as much as possible without altering their underlying configuration, constraining the magnitudes of annual surface NOx emissions, lightning NO, and isoprene emissions as outlined in the paper (lines 105–107) and using consistent fields of $CH_4$ (1760 ppb) under present-day climate. This mimics the conditions that are typically applied in model intercomparisons without greatly perturbing the characteristic behaviour of the models. However, differences between tropical ozone columns, and many other variables, are indeed likely, as they are in model intercomparison studies. As suggested, we have therefore included a direct comparison of baseline model conditions in a supplement to provide the reader with greater insight into the underlying differences between the models.

In general there is a broad consistency between them, although surface NOx and column NO$_2$ are slightly lower and PAN higher in the GISS GCM. The annual global average total column ozone is somewhat less in the GISS GCM (294 vs 321 DU in FRSGC/UCI CTM), but this is dominated by smaller columns at mid and high latitudes, and the tropical columns are very similar (263 vs 265 DU). We now refer to these underlying differences in the discussion, and refer the reader to the table and figures in the supplement for further details. We show examples of this comparison for ozone column and humidity in Figure 3 below.

[Figure]

**Fig 3: Annual mean total column ozone from the FRSGC/UCI CTM, GISS GCM and CAM-Chem (top row) and specific humidity at 500 hPa in July from each model (bottom row).**

*4. The paper does not really pinpoint some of the basic causes of the discrepancies between the models, in particular the different lifetimes of methane. As stated in the previous comment some of the mean fields in the models should be given, for example the global burden of CH$_4$, atmospheric water vapor (perhaps with latitude?), etc. While the sensitivity tests between the different models is revealing evaluating the mean difference between the GISS-GCM and the other models may also be revealing. In a number of locations the authors hypothesize that the difference is due to differences in humidity (lines 177-178, Pages 8 and Page9) or due to differences in the formation/decomposition of PAN. Why dont the authors check? Im not suggesting a lengthy analysis, but an inspection and comparison of some of the mean fields should be sufficient to check some of these hypothesis and possibly reveal some of the key model differences.*

A deeper analysis of a much wider range of variables is needed to really pinpoint the origin of the different behaviour observed, including chemical processes in particular, as noted in the paper. However, the reviewer makes a valuable point that demonstrating the mean state of the different models is relatively straightforward. To address this, we have included surface and mid-tropospheric distributions of a range of trace gases, along with annual mean columns of ozone, NO$_2$ and CO in the supplementary material, and refer to these in the text. With the exception of differences in NOx and PAN, the models are reasonably consistent, suggesting that a deeper analysis of chemistry and transport processes is needed to diagnose

the different responses in OH.

*Minor Comments*

*1. Please state in the caption to Figure 1 that these results are over different years with different emissions and meteorologies. Also please reference Table 1 in the Figure caption where the references to the points can be found. Finally, please state what the dots refer to in the right hand panel.*

The caption has been amended as requested to state "Note that results in the left panel represent differing emissions and meteorological years (study details are given in Table 1)". The dots in the right hand panel are simulations at intermediate points across the ranges specified in the text, but these are not important for understanding the overall responses and they have thus been removed.

*2. The methodology behind the Gaussian process elimination should be explained in more detail in methodology section. While I dont expect the authors to go into detail, neither should it be necessary to reference the referenced papers to understand this analysis.*

We have now added a brief introduction to Gaussian Process emulation, as requested, in Section 2.

*3. Does the sensitivity to boundary layer mixing simply involve changing the vertical diffusion coefficient over the boundary layer depth, or something else?*

Yes, this involves scaling the vertical diffusion coefficient through the depth of the diagnosed boundary layer. The text describing this in Section 2 has been revised to make this clearer: "by scaling the effective vertical diffusion coefficient through the depth of the boundary layer".

*4. The paper states: The models differ in their sources of meteorology, but certainly the sources of meteorology are important. Are the runs using specified dynamics or GCM generated meteorology? This might be quite important to how well the relative humidity is specified. In addition, using a single year might also introduce significant variability between the models. Please say something more specific about the meteorology used as well as comment on the possible importance of the interannual variability. My guess is the latter might introduce significant variability between the models.*

Differences in meteorology are likely to be important, but it is impractical to prescribe a particular source of meteorology given the range of models used. We aimed for consistency with 2001 conditions, but substantial differences are likely to remain. Offline meterological fields from the ECMWF IFS for 2001 were used for the FRSGC/UCI CTM, while the GISS GCM used observed sea-surface temperatures and was nudged to NCEP reanalysis fields. This information was omitted from the description in Section 2, and has now been added.

Interannual meteorological variability is likely to contribute some variability in ozone and OH (Wild et al 2007 found variations of ozone burden of about 2% and of methane lifetime of about 5% over three years), but this is likely to be less that that between entirely different sources of meteorology, where different underlying biases may be present. We have now

revised the text in Section 4 to acknowledge the importance of the meteorological fields, and to note that these are one component of the structural uncertainty associated with the use of different models.

*5. Page 5, line 15. but we scale. It is not altogether clear how this scaling is used. Please specify. Given the non-linear response to NOx, does a linear scaling make sense?*

We apply a simple scaling to the emissions to ensure that the total magnitude is the same in each model. There are differences in the location and timing of emissions, particularly for lightning NO and biogenic isoprene where emission are determined interactively based on model meteorology and vegetation schemes. It is impractical to harmonize these between models, and the differences are effectively structural, as noted in the text. The differences are relatively small, as is evident from the surface distributions of NOx and isoprene that we have now included in the supplement.

*6. Page 7, line 139-141 discrepancies highlight that uncertainty in chemistry and transport processes not considered here may play a substantial role in governing the $CH_4$ lifetime. Im not sure I understand this conclusion. It seems it might simply mean, for example, that the variability in water vapor is considerably over-estimated.*

Reducing the uncertainty in water vapour narrows the distribution, but does not bring the mean lifetime closer to the observed range. The discrepancy between the model and observed values must therefore be due to uncertainty in processes and variables that have not considered in this study. Chemistry and transport processes are likely to be important here, along with other key variables such as the total column ozone. The text has been revised to make this point clearer.

*7. Page 7, line 149 it is notable that humidity has not been prescribed in previous model intercomparison studies. Possibly, but again the large response in water vapor is dependent on the large variation in water vapor.*

The contribution of water vapour is smaller now that the uncertainty range has been reduced to 20%, but it still makes a sizable contribution to the overall uncertainty, as noted in the text here.

*8. Page 8. The explanation in the future responses of the different models is nice and given in several places in the paper. Personally, I think this might belong better in the conclusion section.*

We now refer to this in the conclusions, although we do not include the quantitative aspect here to avoid any further repetition.

*9. Page 9, line 180 suggesting a saturation in OH formation in this model. How does a saturation occur?*

It is unclear why OH is relatively insensitive to water vapour in this model, and it would certainly be useful to explore this further with more targetted studies. Any additional formation from higher water vapour must be balanced by additional removal, but the pathway for this remains unclear. The statement here was intended to indicate that net formation

had reached a plateau, but this has now been rephrased to make it clearer: "suggesting either little additional formation of OH or a matching OH sink in this model".

*10. More generally, I am curious about the conceptual difference between the sensitivity to forcing parameters (NOX emissions, stratospheric ozone column) and internal model parameters. To what extent does it make sense to distinguish between the sensitivity between these type of parameters?*

The approaches used here can be applied to uncertainty in model inputs and internal parameters, and there is no conceptual difference. We make the distinction in the paper to highlight that they can be applied to both, even though previous applications in other contexts have tended to focus on one or the other.

**Response to Reviewer 3:**

*This study is an important contribution towards understanding the reasons for model diversity in the evolution of tropospheric ozone and methane lifetime. However, I found that the approach applied here is inadequate to truly understand the reasons for diversity in these non-linear quantities. My main concern (similar to the other two reviewers) is that the sensitivities calculated for each model would depend on the control simulation given the non-linear chemistry of ozone and methane. If the models differ in the forcings (meteorology and emissions) to begin with then how do we know that the calculated sensitivity is not due to the initial state. Also, I found the description of the Gaussian emulation approach applied here rather limited to appreciate its usefulness for understanding the reasons for diversity in model response.*

We thank the reviewer for these comments, which parallel those of the other reviewers, and we have addressed the principal concerns here in our comments above. We standardised the models in terms of the magnitude of key ozone precursor emissions and in methane burden, and the baseline results of the models (which we have now included in a supplement) are broadly similar. We have revised the text to include a brief introduction to Gaussian Process emulation so that readers unfamiliar with it can appreciate the approach, particularly for representing non-linear responses, which present no problem.

*L23–25: Clarify if this is referring to controls for climate or air pollution. Controls on NOx emissions in the US (e.g., Clean Air Act) and Europe (e.g., LRTAP) have indeed brought down surface ozone.*

As pollutants and greenhouse gases these short-lived gases are attractive targets for both types of control. However, selecting appropriate controls is challenging; while emission controls have reduced surface ozone in the US, similar controls have led to increases in urban ozone in China. The Science paper cited here targets climate change, air quality and food security, and all three areas are of relevance here.

*L27: Also another ACCMIP paper (Naik et al., 2013) and CCMI models (Zhao et al., 2019 https://www.atmos-chem-phys-discuss.net/acp-2019-281/)*

The Naik et al paper is very useful, but covers the same ACCMIP models described in the study of Voulgarakis et al. that we cite. The Zhao et al., study includes valuable new data

from CCMI studies, but methane lifetimes are only available from a limited subset of the models, most of which also contributed to the ACCMIP studies.

*L32: There are large uncertainties in PI estimates of surface ozone as discussed by Tarasick et al. (2019) https://www.elementascience.org/articles/10.1525/elementa.376/*

As noted above in response to reviewer 1, we have rephrased this statement to acknowledge the uncertainties involved and have cited the Tarasick paper which highlights the magnitude of these uncertainties very nicely.

*L58: Observational estimates of global ozone are now available from satellites (Gaudel et al. 2019 https://www.elementascience.org/articles/10.1525/elementa.291/). How do the model estimates discussed here compare with satellite estimates?*

Satellite data provides a useful additional constraint on the global ozone burden, but most assessments are limited to the $60^oS$–$60^oN$ latitude range. Only two IASI retrievals are available for the full globe (333 and 345 Tg), although these exclude regions in polar night. The TOST ozonesone product provides a global burden of 337 Tg, but this is very similar to the ozonesonde-based estimates we already cite. We add an additional sentence acknowledging these additional estimates: "Recent satellite and ozonesonde-based estimates of the global burden range from 333–345 Tg (Gaudel et al., 2019)."

*L100–104: It would help to know how different the base state is in the models. What is the ozone burden, prescribed methane concentration, methane lifetime, surface and lightning NOx emissions, biogenic emissions, wet and dry deposition rates for all species, atmospheric humidity, cloud optical depth, and boundary layer height in the base simulation for all the three models?*

We have now added a direct comparison of the base state of the models in the supplementary material, and provide a very brief summary in the text. As noted in the paper, the magnitude of surface and lightning NO emissions and biogenic emissions are already constrained, as is the methane concentration, but we have included surface and mid-tropospheric distributions of a range of trace gases, along with annual mean columns of ozone, $NO_2$ and CO in the supplementary material to provide a simple characterization of the models.

*L114–115: How are the emulators built for a non-linear system such the $O_3$-NOx-$CH_4$ chemistry? Some description is needed to make the design of emulators transparent for the purpose of this figure.*

We have added a few sentences to provide more details on the method used to generate the emulators. The ability to fully reproduce non-linear behaviour is one of the principal advantages of Gaussian Process emulation, and we now note this in the text.

*L142–143: It would be helpful to provide an equation to explain how sensitivity for each variable is determined. As it stands, the process appears too opaque to me.*

Variance decomposition is performed through calculation of the sensitivity indices using the Sobol approach as described by Saltelli 2002. The mathematical foundation for this is fully described in Ryan et al., 2018, and the code used to generate it is provided at

https://doi.org/10.5281/zenodo.1038667 We have added additional text here to provide a clearer description of the approach taken, and refer the reader to this earlier paper for the numerical theory, notation and equations which underpin it.

Saltelli, A.: Making best use of model evaluations to compute sensitivity indices, Comput. Phys. Commun., 145, 280297, 2002.

*L151: How different is humidity across the three models for the base run? Is it possible that the three models show large sensitivity of ozone to humidity because such a large ($\pm 50\%$) perturbation is used? How do the sensitivities for ozone calculated here compare with those calculated by Revell et al. (2018) https://www.atmos-chem-phys.net/18/16155/2018/acp-18-16155-2018.pdf?*

We have added a simple comparison of summertime humidity distributions in the supplement. The patterns of humidity look very similar in the models, although the magnitudes are slightly less in the GISS GCM. The mean mass of water vapour in the GISS GCM runs ($1.21 \times 10^{16}$ kg) is about 4% less than than in the FRSGC/UCI CTM runs ($1.26 \times 10^{16}$ kg) on an annual basis, and we have included this in a table in the supplement. Revell et al. 2018 did not explicitly consider sensitivity of ozone burden to humidity, but this is a valuable paper and we cite it as another example of use of emulation to explore model sensitivity.

*L167–169: and four models showing decreased lifetime - is the implication here that these four models may have greater sensitivity to humidity and therefore show declining methane lifetime? If so, how do we know that these models are like CAM-chem and FRSGC/UCI CTM in their sensitivities?*

Greater future humidity would suggest greater OH formation and hence reduced methane lifetime, although this is balanced against reduced secondary OH formation if NOx emissions are reduced. The point we make here is that the relative sensitivity to water vapour and NOx emissions is important, and that the different model behaviours are very likely associated with differing sensitivities. We cannot explain the behaviour of these other ACCMIP models without further analysis, but suggest that exploring sensitivities with them would be very revealing.

*L192–194: The chemical loss of methane also depends on the concentration of methane in the models. Are they the same across the models?*

We constrain methane to a fixed concentration of 1760 ppb suitable for 2001 conditions, so that methane is the same in all models. We neglected to highlight this in the text, and have now amended the discussion in Section 2 to include this.

*L198: How different are the model chemical mechanisms implemented in the models? I would imagine the differences in sensitivities due to NOx are due to the implemented chemical mechanisms.*

The chemical mechanisms in the models differ, and this is one set of processes that we were not able to explore (as noted on lines 203–204), as it requires consideration of a much larger number of variables than we were able accommodate (currently just eight). Previous studies have explored the contribution of chemical processes in a single model (e.g., Newsome and

Evans, 2017, as cited in the paper), and further exploration of this across different models using the approaches described here would be a valuable extension of this study, as we note in the Conclusions.